# Honokiol and Nicotinamide Adenine Dinucleotide Improve Exercise Endurance in Pulmonary Hypertensive Rats Through Increasing SIRT3 Function in Skeletal Muscle

**DOI:** 10.3390/ijms252111600

**Published:** 2024-10-29

**Authors:** Min Li, Brittany Alexandre McKeon, Sue Gu, Ram Raj Prasad, Hui Zhang, Sushil Kumar, Suzette Riddle, David C. Irwin, Kurt R. Stenmark

**Affiliations:** Cardiovascular Pulmonary Research Laboratories, Departments of Pediatrics and Medicine, School of Medicine, University of Colorado Anschutz Medical Campus, Aurora, CO 80045, USA

**Keywords:** pulmonary hypertension, SIRT3, Honokiol, NAD, mitochondrial, skeletal muscle, exercise, critical speed

## Abstract

Pulmonary hypertension (PH) significantly impairs exercise capacity and the quality of life in patients, which is influenced by dysfunctions in multiple organ systems, including the right ventricle, lungs, and skeletal muscles. Recent research has identified metabolic reprogramming and mitochondrial dysfunction as contributing factors to reduced exercise tolerance in PH patients. In this study, we investigated the therapeutic potential of enhancing mitochondrial function through the activation of the mitochondrial deacetylase SIRT3, using SIRT3 activator Honokiol combined with the SIRT3 co-factor nicotinamide adenine dinucleotide (NAD), in a Sugen/Hypoxia-induced PH rat model. Our results show that Sugen/Hypoxia-induced PH significantly impairs RV, lung, and skeletal muscle function, leading to reduced exercise capacity. Treatment with Honokiol and NAD notably improved exercise endurance, primarily by restoring SIRT3 levels in skeletal muscles, reducing proteolysis and atrophy in the gastrocnemius, and enhancing mitochondrial complex I levels in the soleus. These effects were independent of changes in cardiopulmonary hemodynamics. We concluded that targeting skeletal muscle dysfunction may be a promising approach to improving exercise capacity and overall quality of life in PH patients.

## 1. Introduction

Pulmonary hypertension (PH) is characterized by pulmonary vascular remodeling, including vascular cell proliferation, apoptosis resistance, inflammation, and fibrosis, that leads to an increased right ventricular (RV) afterload, ultimately culminating in RV failure and death [1,2]. Five distinct categories of PH have been identified on the basis of clinical parameters, potential etiologic mechanisms, and pathophysiological characteristics [3,4]. PH is a progressive and debilitating disease due to a variety of pathologic insults with no cure, and the median survival time is approximately 5–7 years [5]. As the disease progresses, pathological changes affect both the physiological and psychological aspects of patients, resulting in various symptoms, the most notable being shortness of breath, fatigue, and reduction in daily life activities [6,7,8,9,10]. Exercise intolerance, therefore, severely compromises patients’ independence and quality of life (QoL).

Physical exercise demands the integrated response of the cardiovascular, pulmonary, and musculoskeletal systems. In PH patients and animal models, RV dysfunction, impaired ventilation, reduced pulmonary circulation, and compromised gas exchange are key mechanisms contributing to reduced exercise capacity [11,12,13,14,15]. Over the past decade, research has advanced our understanding of PH as a complex clinical syndrome that affects multiple organ systems. The reduction in exercise capacity in PH patients is not solely due to central cardiopulmonary impairments; skeletal muscle structural and functional abnormalities also play a critical role in regulating exercise capacity [7,16,17,18,19]. Although the underlying mechanisms of skeletal muscle dysfunction in PAH remain incompletely understood, the impact of these abnormalities on patients’ quality of life is undeniable. In addition, despite numerous advances in medical therapy that have notably enhanced patient survival, decreased exercise tolerance remains a predominant symptom experienced by patients [7,20]. Therefore, investigations aimed at elucidating these mechanisms, as well as therapies capable of reversing or mitigating skeletal muscle damage, are of significant clinical interest.

Emerging evidence indicates that PAH is a systemic metabolic disease [21,22,23,24,25,26]. Metabolic reprogramming and mitochondrial dysfunction are observed not only in the lungs, pulmonary vasculature, and RV but also in the skeletal muscles of patients [27,28,29]. For example, proteomic and metabolic analysis of skeletal muscle from idiopathic PAH (IPAH) patients have revealed abnormal mitochondrial morphology, downregulation of proteins closely associated with mitochondrial structure and function, and a metabolic shift in the quadriceps from oxidative metabolism to anaerobic glycolysis [27]. In the Sugen/Hypoxia-induced PH rat model, significant decreases in exercise endurance were linked to a downregulated transcriptional profile related to mitochondrial function [30]. Thus, metabolic and mitochondrial abnormalities may represent a common mechanism underlying both central cardiopulmonary and peripheral musculoskeletal impairments that contribute to exercise intolerance. SIRT3 is a mitochondrial deacetylase and requires nicotinamide adenine dinucleotide (NAD^+^) as a cofactor for its enzymatic activity. Decreased SIRT3 activity has been reported in various disease conditions, including PH [31,32,33]. Honokiol is the best-recognized SIRT3 agonist and a natural pleiotropic lignan found in the plant of Magnolia grandiflora and is demonstrated as a potent anticancer, neuroprotective, and antioxidant agent [34,35,36]. In a previous study, we demonstrated that treatment of pulmonary artery adventitial fibroblasts isolated from IPAH patients and hypoxia-induced PH bovines (PH-Fibs) with a combination of the SIRT3 activator Honokiol and the SIRT3 co-factor nicotinamide adenine dinucleotide (NAD^+^) improved mitochondrial function and inhibited the proliferation of PH-Fibs [32]. However, it remains unknown whether a systemic response involving the RV, lung, and skeletal muscle can be achieved with this treatment to enhance SIRT3 activity, restore mitochondrial function, and improve exercise capability in PH.

In this study, we aim to examine the protein levels of mitochondrial deacetylase SIRT3 in the heart, lung, and skeletal muscle, evaluate cardiopulmonary function, and exercise endurance using a Sugen/hypoxia-induced PH rat model. We also investigated the mechanisms by which SIRT3 restoration, using the combination of SIRT3 activator Honokiol and SIRT3 co-factor nicotinamide adenine dinucleotide (NAD), affects exercise capability. We hypothesize that increasing SIRT3 activity can improve cardiopulmonary and skeletal muscle function by restoring mitochondrial function, thus enhancing exercise endurance in PH.

## 2. Results

### 2.1. Sugen/Hypoxia Deteriorates and Honokiol/NAD Treatment Improves Exercise Capacity in PH Rats

To evaluate the effects of Honokiol/NAD treatment on exercise intolerance in PH, male Sprague-Dawley rats at 8 weeks of age were administered Sugen 5416 and exposed to hypoxia for 3 weeks, followed by a return to normoxic conditions at sea level for an additional 3 weeks. During the 6-week period, a subset of the Sugen/hypoxia rats received daily doses of the mitochondrial deacetylase SIRT3 activator, Honokiol, and the SIRT3 co-factor Nicotinic Acid (NA, a precursor of NAD). The control group of rats was kept in sea-level normoxic conditions for 6 weeks. At the end of the 6 weeks, a critical speed (CS) test of exercise capacity was conducted (Figure 1) according to previously published methods [37,38,39]. Results indicated that rats in the Sugen/hypoxia-induced PH group (SH) exhibited a significant reduction in CS compared to the normoxic control group (NX) (*** *p* < 0.001, **** *p* < 0.0001), reflecting diminished exercise endurance. Treatment with Honokiol and NAD (SH/HN) significantly improved CS in the SH rats (# *p* < 0.05 compared to SH rats), although it remained decreased compared to NX rats (* *p* < 0.05) (Figure 2). The CS data were analyzed using three distinct models as described in the methods to ensure accuracy and avoid bias, with representative results presented in Appendix A. Any results with a coefficient of determination (R^2^) less than 0.95 were excluded.

### 2.2. Sugen/Hypoxia Decreased SIRT3 Protein Levels in Lung, RV, and Skeletal Muscles; Honokiol and NAD Treatment Restored SIRT3 Levels Only in Skeletal Muscles

To investigate the mechanisms underlying the restorative effects of Honokiol/NAD on exercise intolerance, we measured SIRT3 protein levels in the lung, RV, and skeletal muscle (gastrocnemius and soleus). Western blot analysis revealed a significant decrease in SIRT3 protein levels in the lung (*p* < 0.01), RV (*p* < 0.05), gastrocnemius (*p* < 0.05), and soleus (*p* < 0.0001) of SH rats compared to NX rats. However, Honokiol and NAD treatment significantly increased SIRT3 protein levels in gastrocnemius and soleus muscles (*p* < 0.01) but did not restore SIRT3 levels in the lung or RV (Figure 3).

### 2.3. Honokiol and NAD Treatment Did Not Decrease Severity of PH

We next assessed hemodynamic parameters to evaluate cardiopulmonary function. Sugen/hypoxia significantly elevated mean pulmonary artery pressure (mPAP), PA pulse pressure, and pulmonary vascular resistance (PVR) while significantly reducing PA compliance (Figure 4), indicating substantial pulmonary vascular remodeling. Additionally, Sugen/hypoxia significantly increased Fulton’s index (RV/LV+S ratio) RV end-systolic and end-diastolic pressure, reflecting adaptive RV hypertrophy and diastolic dysfunction. However, RV cardiac output was maintained, and RV contractility was significantly increased by pressure recruitable stroke work (PRSW) slope with a trend towards improvement by end-systolic elastance (Ees), without a decrease in RV/PA coupling (the ratio of end-systolic elastance to arterial elastance, Ees/Ea) (Figure 4). Treatment with Honokiol and NAD did not alter pulmonary artery pressure or resistance, while there was a trend towards an increase in RV contractility (by Ees and PRSW slope) and improvement in RV/PA coupling (Figure 4).

### 2.4. Honokiol and NAD Affected Certain Metabolic and Inflammatory Gene Expression in RV of Sugen/Hypoxia Rats

We further examined the impact of Honokiol and NAD treatment on the expression of genes critical to cardiopulmonary function. The treatment significantly reduced the expression of metabolic genes (*Glut1*, *Ldha*) and certain inflammatory genes (*Sdf1*, *Vcam1*) in the RV, although it did not affect the expression of genes related to proliferation and remodeling, except for a significant reduction in fibronectin expression (Figure 5). The treatment did not significantly affect the expression of lung genes that were tested.

### 2.5. Sugen/Hypoxia Increased Proteolysis and Muscle Atrophy in Gastrocnemius of PH Rats; Honokiol/NAD Treatment Decreased Proteolysis Gene Expression and Increased Muscle Mass

We analyzed skeletal muscle mass and found that Sugen/hypoxia significantly reduced muscle mass in the gastrocnemius but not in the soleus (Figure 6A). Real-time RT-PCR analysis of proteolysis-related gene expression revealed a significant increase in the expression of *Murf1*, *Foxo3a*, and *Cathespin-l* in the gastrocnemius but not in the soleus. While *Atrogin-1* and *Foxo1* expression showed an increasing trend in the gastrocnemius, they did not reach statistical significance (Figure 6B). Honokiol and NAD treatment significantly decreased the expression of *Murf1*, *Foxo3a*, *Cathespin-l*, and *Atrogin-1* and showed a decreasing trend for *Foxo1* in the gastrocnemius (Figure 6B).

### 2.6. Honokiol and NAD Treatment Increased Mitochondrial Complex I Level of Skeletal Muscle in Sugen/Hypoxia PH Rats

To evaluate the effects of SIRT3 restoration on mitochondrial function in skeletal muscle, we measured mitochondrial complex protein levels. Both mitochondrial complexes I and II were significantly downregulated in the gastrocnemius and soleus muscles of Sugen/hypoxia PH rats. Honokiol and NAD treatment either significantly (complex II in gastrocnemius and complex I in soleus) or showed an increasing trend (complex I in the gastrocnemius and complex II in soleus) of mitochondrial complex protein expressions in both muscle types, although some changes did not reach statistical significance (Figure 7). Additionally, glycolytic gene expression analysis revealed that *Hk2* and *Ldha* gene expression was significantly increased in the gastrocnemius, but not the soleus, of SH rats. Honokiol and NAD treatment significantly decreased *Hk2* and *Ldha* gene expression in the gastrocnemius (Figure 8).

### 2.7. Nicotinamide Adenine Dinucleotide (NAD) Concentrations in Skeletal Muscles Are Decreased in Sugen/Hypoxia PH Rats; Honokiol and NAD Treatment Did Not Fully Restore NAD Levels

Given that SIRT3 requires nicotinamide adenine dinucleotide as a co-factor to perform its mitochondrial deacetylase function, we measured NAD concentrations in skeletal muscles. We found that NAD concentrations are significantly decreased in both gastrocnemius and soleus muscles of Sugen/hypoxia PH rats. While Honokiol and NAD treatment increased NAD concentration, the change did not reach statistical significance (Figure 9)

## 3. Discussion

In this study, we investigated whether enhancing mitochondrial function with the use of the mitochondrial deacetylase SIRT3 activator Honokiol, in combination with the SIRT3 co-factor nicotinamide adenine dinucleotide (NAD), could improve cardiopulmonary and skeletal muscle function, thereby increasing exercise capacity in a Sugen/Hypoxia-induced PH (SH) rat model. Our findings demonstrate that Sugen/Hypoxia-induced PH significantly impairs RV, lung, and skeletal muscle function, leading to exercise intolerance in these SH rats compared to controls. Treatment with Honokiol and NAD improved exercise capacity in PH rats, primarily by increasing SIRT3 levels in the gastrocnemius and soleus muscles, reducing proteolysis and atrophy in the gastrocnemius, and increasing mitochondrial complex protein levels in the gastrocnemius and soleus. Although treatment reduced the expression of certain inflammation-related (*Sdf1*, *Vcam1*) and metabolism-related (*Glut1* and *Ldha*) genes in the RV, pulmonary artery pressure and resistance remained unchanged. These results suggest that the improvement in exercise capacity is driven primarily by the peripheral skeletal muscle response to Honokiol and NAD treatment rather than central cardiopulmonary improvements. The implications of these findings are that in the context of PAH, treatments or interventions targeting SIRT3 and mitochondrial function in skeletal muscles could lead to improvements in functional status, as indicated by increased 6MWD, potentially enhancing the quality of life for the patient. However, this peripheral response appears to be independent of central cardiopulmonary system changes and does not always correlate with hemodynamic improvements.

Two different skeletal muscle types, namely soleus and gastrocnemius, were examined in this study. The soleus is predominantly composed of type I slow-twitch oxidative fibers (~80%), while the gastrocnemius contains more type II fast-twitch glycolytic fibers. Type I fibers possess endurance-oriented properties and exhibit higher mitochondrial content, higher oxidative capacity, better maintained respiratory function (especially with aging), and more active mitochondrial dynamics compared to the type II fibers; thus, type I fibers are more fatigue- and atrophy-resistant. Conversely, type II fibers can be characterized as power-oriented and involved in explosive movements. Type II fibers are more glycolytic, fatigued easily, and prone to atrophy. This fundamental difference in fiber type composition suggests inherent differences in metabolic activity, mitochondrial function, and structural and functional alteration in response to disuse, aging, and disease conditions [40,41,42]. Muscle wasting is a common feature in conditions such as advanced diabetes, aging, hypoxia-related pathologies, and cancer cachexia [43,44,45,46]. Our data demonstrated gastrocnemius atrophy in Sugen/Hypoxia-induced PH rats, consistent with findings that muscle atrophy is present in both PH human patients and animal models, as well as human skeletal muscles response to high altitude [30,47,48,49,50], along with declined muscle strength and exercise capacity [30,45,51,52]. For instance, severe muscle atrophy and an associated anabolic/catabolic imbalance have been observed in the gastrocnemius of monocrotaline (MCT)-induced PAH rats [53]. Additionally, the cross-sectional area (CSA) and maximal force of quadriceps fibers are significantly reduced in PAH patients [54]. Muscle mass maintenance is determined by the balance between protein synthesis and degradation. In this study, we identified upregulation of the prototrophic ubiquitin ligases atrogin-1 and Muscle RING finger protein-1 (*MuRF1*), as well as the transcriptional factor Forkhead box (FOX) O (*FoxO3a*), which is shown to be a key regulator of the muscle atrophy pathway [55,56]. The novel finding of this study is that Honokiol and NAD treatment significantly decreased the expression of these genes and reduced gastrocnemius atrophy.

SIRT3, a member of the sirtuin family, requires NAD^+^ as a co-factor for its enzymatic activity. As the primary mitochondrial deacetylase, SIRT3 decreases acetylation levels and thereby increases the activity of numerous mitochondrial proteins [57,58]. Emerging evidence suggests a role for SIRT3 in the pathogenesis of PH. Decreased SIRT3 levels and activity have been demonstrated in pulmonary artery smooth muscle cells (PASMC) of PAH human patients and MCT-induced PH rats [33], as well as in lung tissues and pulmonary artery adventitial fibroblasts of PH bovine model and human patients [32]. In a different subtype of PH, associated with heart failure with preserved ejection fraction (PH-HFpEF), chronic oral nitrite treatment has been shown to potentially prevent disease progression through activation of the SIRT3- AMPK-GLUT4 signaling pathway in skeletal muscle, but not in the lung, LV, RV, or liver [59]. Interestingly, the same group recently demonstrated a systemic pathogenic impact of skeletal muscle-specific SIRT3 deficiency on remote pulmonary vascular remodeling and PH-HFpEF [60]. SIRT3 has been reported to regulate FoxO3a function in endothelial cells by modulating its acetylation level [61]. In knee osteoarthritis, SIRT3 regulates Atrogin-1 and MuRF-1 levels by decreasing SOD2 acetylation and reducing reactive oxygen species (ROS) levels [62]. However, the mechanisms by which SIRT3 regulates the expression of these gene expressions in the context of PH remain unclear and need further investigation. Hypoxic conditions, impaired pulmonary circulation, reduced capillary density and oxygen delivery to muscles, as well as decreased mitochondrial efficiency in utilizing oxygen, all contribute to skeletal muscle hypoxemia and dysfunction [7,16,48,63]. Whether Honokiol and NAD influence angiogenesis and oxygen delivery to skeletal muscle also remains to be further investigated.

In this study, we observed inconsistent relationships between exercise endurance and hemodynamic parameters following the treatment. While improving cardiopulmonary function is critical for managing PAH and enhancing exercise tolerance, the link between exercise endurance and hemodynamics remains debated. For instance, a cohort study of patients with inoperable chronic thromboembolic pulmonary hypertension (CTEPH) who had undergone balloon pulmonary angioplasty (BPA) found that 55.9% of patients with a higher physical component summary (PCS) score after BPA did not achieve the goal of mean pulmonary arterial pressure (mPAP) of ≤30 mmHg, despite improved physical function [64]. Additionally, while muscle strength is decreased in PAH patients and 6MWD correlates significantly with handgrip strength, no correlation was found between muscle strength and N-terminal pro-brain natriuretic peptide levels, an indicator of heart function [52]. Furthermore, patients with a persistently low 6MWD, despite improved hemodynamics, often have poor prognosis [65]. This reflects the complexity of the disease and variability in treatment response, with non-hemodynamic factors, such as skeletal muscle function, emerging as potential therapeutic targets.

In conclusion, this is the first study to evaluate the effect of Honokiol and NAD on exercise endurance in a PH rat model. We provide evidence that in Sugen/Hypoxia-induced PH rats, there is a decrease in mitochondrial deacetylase SIRT3 protein levels and a reduction in the concentration of the SIRT3 co-factor NAD in both gastrocnemius and soleus muscles. Skeletal muscle function is impaired, as evidenced by increased proteolysis, gastrocnemius atrophy, mitochondrial dysfunction, and significant exercise intolerance. The combination treatment of the SIRT3 activator Honokiol and the NAD precursor nicotinic acid increased SIRT3 protein levels in skeletal muscles, reduced proteolysis and gastrocnemius atrophy, and increased mitochondrial protein level, leading to improved exercise capacity in PH rats.

## 4. Materials and Methods

### 4.1. Hypoxia Exposure, Critical Speed Test and Treatment Strategy

Male Sprague-Dawley rats (7 weeks old) were obtained from Charles River Lab (Wilmington, MA, USA). Rats were kept in sea-level chambers on the day they were delivered for 2 weeks and housed in an AAALAC-accredited animal facility at the University of Colorado, Denver, Anschutz Medical campus. During the second week, all rats under sea-level conditions received treadmill familiarization training. Rats were maintained on a 12:12-h light-dark cycle with food and water available ad libitum. Rats were then randomly assigned into three groups: (1) NX: normoxia group (n = 8 total). Rats were kept in sea level normoxic condition for 6 weeks. (2) SH: Sugen/hypoxia group (n = 9 total). Rats were given one single injection of SU5416 (20 mg/kg, i.p.) and placed in hypobaric hypoxic chambers (0.5 atm) with a simulated altitude of 18,000 ft in a ventilated chamber for 3 weeks. The chamber was flushed with a mixture of room air and nitrogen, and the gas was re-circulated. The hypoxic condition was monitored using an oxygen monitor. Rats were moved back to sea level normoxic chamber after 3 weeks’ hypoxia exposure and kept at sea level for another 3 weeks. (3) SH/HN: Sugen/hypoxia plus the combination of Honokiol and Nicotinic acid treatment group (n = 6 total). Rats were given a single SU5416 injection and kept in a hypoxia chamber for three weeks, then moved back to sea level for another 3 weeks. During the whole 6-week period, rats were treated with SIRT3 activator Honokiol (40 mg/kg, gavage, daily) and nicotinic acid (8.5 mg/100 g, gavage, daily), a precursor of SIRT3 co-factor nicotinamide adenine dinucleotide (NAD^+^). At the end of the 6-week, all rats completed critical speed tests (Figure 1).

### 4.2. Treadmill Exercise and Constant Speed Tests

All rats completed a treadmill familiarization phase before the critical speed (CS) test. Familiarization consists of five ~5-min runs on a motor-driven rodent treadmill (Exer 3/6, Columbus Instruments, Columbus, OH, USA). For the first several runs, the treadmill speed was maintained at 20 m/min (up a 5° grade, which was maintained throughout all treadmill tests). For the last several runs, the speed of the treadmill increased progressively to ~40–50 m/min to familiarize the rats with high-speed running. Animals were encouraged to run with intermittent bursts of compressed room air aimed at the hind limbs from directly above the animal (so as not to push the rat up the treadmill). All treadmill testing protocols were designed and conducted by experienced staff and strictly followed the guidelines set by the American Physiological Society’s resource book for the design of animal exercise protocols. Rats presented no pain or discomfort associated with treadmill exercise and conducted healthy eating, drinking, and grooming activities while housed. All experimental procedures were approved by the Institutional Animal Care and Use Committee at the University of Colorado, Denver, Anschutz Medical Campus. Protocol code 252 and the approval date is 29 November 2021.

The CS was determined using a modified version of the methodology used by Copp and Poole et al. [39]. After completion of the treadmill familiarization period, each rat will perform 3–5 runs to exhaustion, in random order, at a constant speed that resulted in fatigue between 1 and 20 min (speeds ranging from 30 to 55 m/min). Each test will be performed on separate days with a minimum of 24 h between tests. For each constant-speed trial, rats will be given a 2-min warm-up period where they ran at 20 m/min followed by a 1-min period of quiet resting. To start the test, the treadmill speed is increased rapidly over a 20-s period to the desired speed, at which point a stopwatch was started. Testing was terminated, and time to exhaustion was measured to the nearest tenth of a second whenever the rats could no longer maintain pace with the treadmill despite obvious exertion of effort. A successful constant-speed test was determined if (1) the rat could quickly adapt to the treadmill speed at the beginning of the test (e.g., did not waste energy), (2) a noticeable change in gait occurred preceding exhaustion (i.e., lowering of the hindlimbs and rising of the snout), and (3) the animal’s righting reflex was markedly attenuated when placed on their back in a supine position (an unexhausted quadruped will typically attempt to right themselves within ~1 s) [38].

### 4.3. Data Modeling for the Determination of CS

Following the successful completion of the constant speed treadmill tests, the CS was calculated for each rat using three models as described previously [37,38,66]: (1) the hyperbolic speed model [Speed= (D’/time) + CS] where the asymptote of the hyperbolic curve is the CS, and the curvature constant is the D’ (finite distance), (2) the linear 1/time model (Speed = D’ × 1/time + CS) whereby the treadmill speed used for the constant speed test is plotted as a function of the inverse of time to fatigue, and the y-intercept of the regression line yields the CS, and the slope is the D’, and (3) the linear distance/time model, where the distance run by the mouse is plotted against the time to fatigue (Distance = CS × time + D’) and the slope of the regression line is the CS and intercept is the D’. CS was calculated using these three models to rule out any tendency for one model to over or underestimate the CS. Any results with a coefficient of determination (R^2^) less than 0.95 were excluded.

### 4.4. Pressure-Volume Measurements

Animals were anesthetized with 4–5% isoflurane in room air at 1 lpm for 2 min; a tracheal cannula was inserted and connected to a Hallowell EMC Anesthesia Workstation ventilator. Anesthesia was maintained at 1.5–2.5% isoflurane in 21% oxygen/ balance nitrogen. Peak airway pressure was maintained between 14 and 18 cmH2O, breaths per minute at 50, and the oxygen flow at 0.5–0.8 lpm. A pressure-only catheter (FTH-1611B-0018, Transonic/Scisense, London, ON, Canada) was placed into the femoral artery for continuous recording of systemic pressure. A 1.9-Fr Pressure-Volume Catheter (Transonic/Scisense, FTH-1912B-6018) catheter was placed into the pulmonary artery via thoracotomy, and pressure measurements were briefly recorded. The RV and LV data were measured with the pressure-volume catheter inserted into the heart through the wall via a diaphragmatic approach. In brief, the pericardium was resected, and a small entry hole was made at the apex of the RV with a 26-gauge needle. The pressure-volume catheter was inserted and centered along the length of the RV. Steady-state hemodynamics were collected with short pauses in ventilation (up to 10 s) to eliminate ventilator artifacts from the pressure-volume recordings. Transient occlusions of the inferior vena cava were also performed to decrease preload and obtain an accurate end-systolic pressure-volume relationship. One final puncture was made in the apex of the LV, and the catheter was centered along the length of the LV to measure steady-state data in the LV. Data were recorded continuously with LabScribe 23 (iWorx, Dover, NH, USA) and analyzed offline.

### 4.5. Real-Time RT-PCR

Total RNA from the rat’s right ventricle, lung, and skeletal muscles (gastrocnemius and soleus) were isolated using the Trizol method according to the manufacturer’s instruction (Qiagen, San Diego, CA, USA). First-strand cDNA was synthesized with iScript cDNA Synthesis Kit (Bio-RAD, Hercules, CA, USA). qPCR reactions were carried out on Applied Biosystems by Life Technologies under the following conditions: initial denaturation at 95 °C for 10 min followed by 40 cycles of denaturation at 95 °C for 15 s and combined annealing and extension at 60 °C for 60 s. HPRT was used as a housekeeping gene. The 2^−∆∆Ct^ method was employed to calculate the fold change of gene expression relative to normoxic controls. The following genes were determined for different tissues: Gastrocnemius and soleus—*Murf1*, *MAFbx (Atrogin-1)*, *Foxo3a*, *Cathepsin-L*, *Foxo1*, *Hk2*, *Ldha*, *Glut1*. Right ventricle—*Glut1*, *Ldha*, *Sdf1*, *Vcam1*, *Il6*, *Mcp1*, *Mki67*, *Cdk1*, *Tnc*, *Fibronectin*, *Nppa*, *Nppb*, *Acta*. The sequences for all primers are listed in Appendix A.

### 4.6. Western Blotting

Primary antibodies used for this study are as follows: SIRT3-specific rabbit monoclonal antibody was purchased from Cell Signaling Technology (Beverly, MA, USA) and used at 1: 1000 dilutions. Mitochondrial complex I (NDUFS8)-specific mouse monoclonal antibody was purchased from Abcam (Boston, MA, USA) and used at 1:1000 dilution. Mitochondrial complex V (ATP5A)-specific mouse monoclonal antibody was purchased from Abcam (Boston, MA, USA) and used at 1:1000 dilution. Mitochondrial complex II (SDHB)-specific mouse monoclonal antibody was purchased from Abcam (Boston, MA, USA) and used at 1:1000 dilution. Beta-actin-specific mouse monoclonal antibody was purchased from Sigma-Aldrich (Burlington, MA, USA) and used at 1:4 k dilution. Equal protein amounts (25 μg of whole cell lysates) from each sample were loaded in sodium dodecyl sulfate-polyacrylamide gel electrophoresis. Gels were transferred, membranes were incubated with antibodies against the proteins described above, and processed as previously described [32]. Quantitative analysis was performed with Image J software (V1.53, ImageJ, U.S. National Institutes of Health, Bethesda, MD, USA) densitometry.

### 4.7. Assessment of NAD^+^ and NADH

The quantification of NAD^+^ concentration in skeletal muscles was determined using an NAD/NADH quantitation colorimetric assay kit (Abcam, Boston, MA, USA) following the instructions.

### 4.8. Statistical Analysis

All statistical analysis was completed using GraphPad Prism version 10.0 (GraphPad Software, San Diego, CA, USA), and significance was set at *p* ≤ 0.05. One-way ANOVA followed by Turkey’s post-test analysis for multiple comparisons was used. Values were expressed as mean ± SEM.

## Figures and Tables

**Figure 1 ijms-25-11600-f001:**
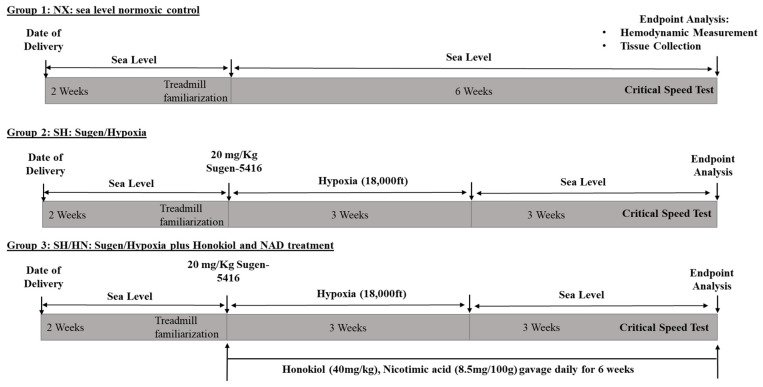
Animal study protocol. Illustration of Sugen/Hypoxia exposure, Honokiol+NAD treatment, and critical speed testing protocol in three groups of rats.

**Figure 2 ijms-25-11600-f002:**
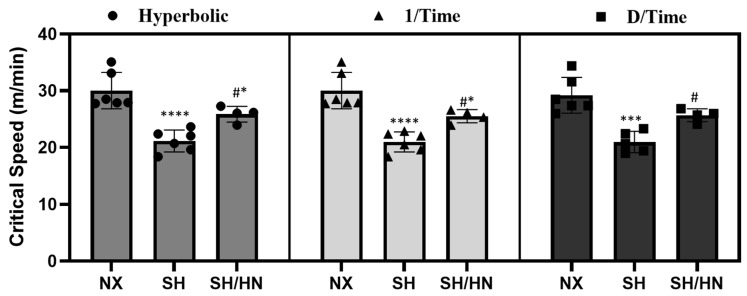
Sugen/Hypoxia decreased and Honokiol+NAD treatment improved exercise endurance in rats. A critical speed test was performed at the end of week 6 for each rat in three different groups: NX sea level normoxia group (6 weeks total). SH, Sugen/Hypoxia (3-weeks hypoxia + 3-weeks sea level normoxia). SH/HN, Sugen/Hypoxia (3-weeks hypoxia + 3-weeks sea level normoxia) plus Honokiol+NAD treatment. The results were analyzed using three models: hyperbolic (time vs. speed), 1/Time (speed vs. 1/time), and D/Time (time vs. distance). * *p* < 0.05, *** *p* < 0.001, **** *p* < 0.0001 compared to normoxic group. #, *p* < 0.05 compared to the Sugen/hypoxia group.

**Figure 3 ijms-25-11600-f003:**
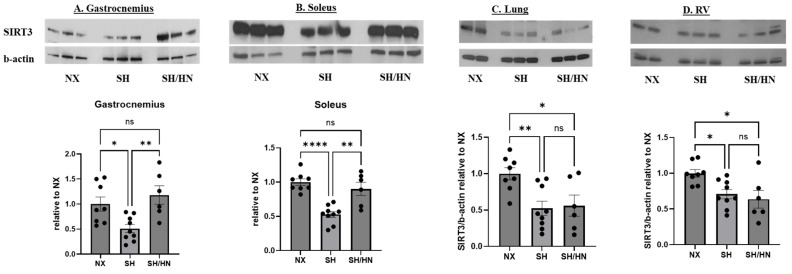
SIRT3 protein levels in lung, RV and skeletal muscles of sea level normoxic, sugen/hypoxic, and honokiol/NAD treated rats. Skeletal muscle ((**A**) gastrocnemius and (**B**) soleus), lung (**C**), and right heart (RV (**D**)) tissues were isolated from sea level normoxic control (NX), Sugen/Hypoxia (SH), and SH plus Honokiol+NAD treated rats (SH/HN). Proteins were isolated from the tissues, and a western blot was performed with antibodies against SIRT3 and beta-actin. The ratio of SIRT3 expression to beta-actin was calculated, and the data was presented as the fold change relative to the NX group. Data was shown as mean ± sem. * *p* < 0.05, ** *p* < 0.01, **** *p* < 0.0001. ns, not statistically different.

**Figure 4 ijms-25-11600-f004:**
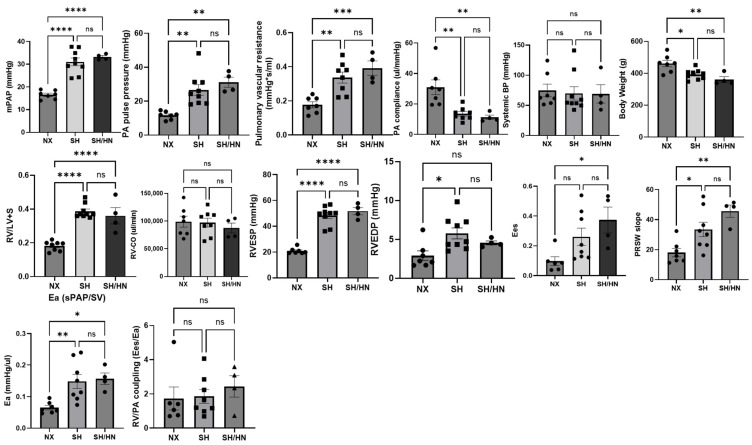
Hemodynamic measurements of sea level normoxic (NX), Sugen/Hypoxic (SH), and Honokiol+NAD treated (SH/HN) rats. * *p* < 0.05, ** *p* < 0.01, *** *p* < 0.005, **** *p* < 0.0001 compared to NX. ns, not statistically different.

**Figure 5 ijms-25-11600-f005:**
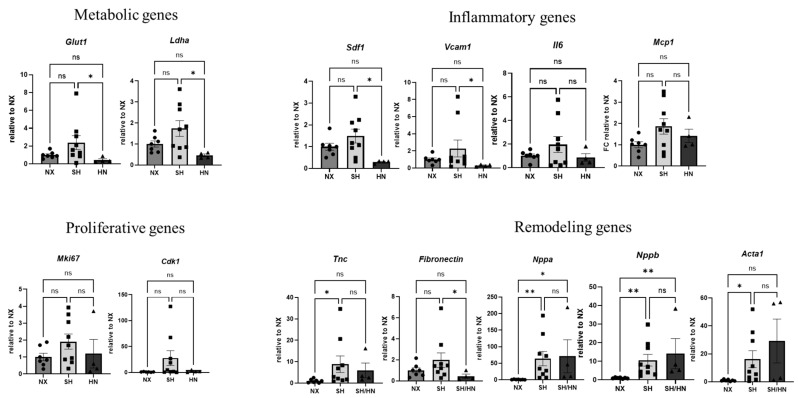
Gene expression of RV tissues of sea level normoxic (NX), Sugen/Hypoxic (SH), and Honokiol+NAD treatd (SH/HN) rats. mRNA was isolated from RV tissues of sea-level normoxic control rats (NX), Sugen/Hypoxia (SH) induced pulmonary hypertensive rats, and SH plus Honokiol+NAD treated rats (SH/HN). Real-time RT-PCR was performed to examine gene expression. Data was calculated as the fold changed relative to NX control and shown as mean ± sem. * *p* < 0.05, ** *p* < 0.01. ns, not statistically different.

**Figure 6 ijms-25-11600-f006:**
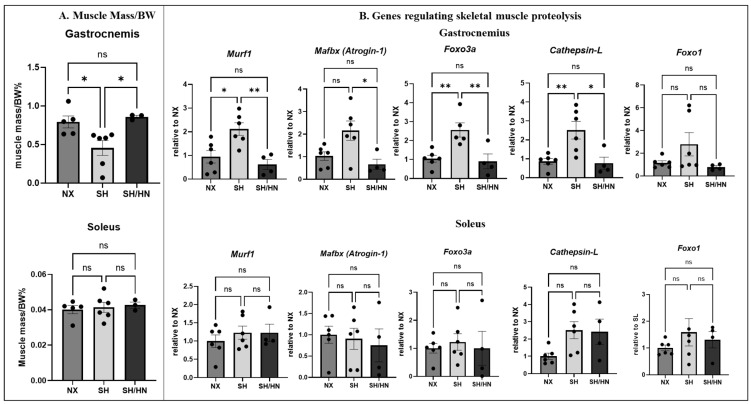
Gastrocnemius muscle mass decreased, and proteolysis gene expression increased in Sugen/Hypoxic PH rats. Honokiol+NAD treatment decreased proteolysis gene expression and increased gastrocnemius mass. Skeletal muscles (gastrocnemius and soleus) were collected from sea-level normoxic control rats (NX), Sugen/Hypoxia (SH) induced pulmonary hypertensive rats, and SH plus Honokiol and NAD-treated rats (SH/HN). (**A**) The lean mass of muscles was measured, and the data was presented as the ratio of muscle mass to body mass and shown as mean ± sem. * *p* < 0.05. (**B**) The mRNA was isolated from the muscles, and proteolytic genes were determined using real-time RT-PCR. The data was presented as fold change relative to the NX group and shown as mean ± sem. * *p* < 0.05, ** *p* < 0.01. ns, not statistically different.

**Figure 7 ijms-25-11600-f007:**
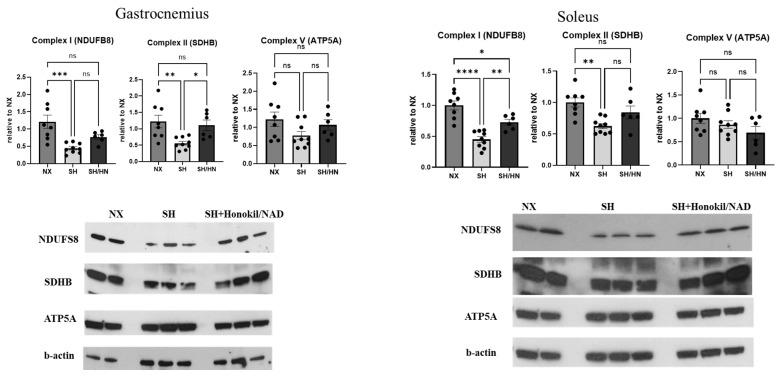
Mitochondrial protein levels are downregulated in the skeletal muscles of Sugen/Hypoxic rats and restored by the treatment of Honokiol and NAD. Skeletal muscle tissues (gastrocnemius and soleus) were isolated from sea level control (NX), Sugen/Hypoxia (SH), and SH plus Honokiol+NAD treated rats (SH/HN). Proteins were isolated from the tissues, and a western blot was performed with an antibody against mitochondrial complex I-NDUFB8, II-SDHB, V-ATP5A, and beta-actin. The ratio of mitochondrial protein expression to beta-actin was calculated, and the data was presented as the fold change relative to the NX group. Data was shown as mean ± sem. * *p* < 0.05, ** *p* < 0.01, *** *p* < 0.005, **** *p* < 0.001. ns, not statistically different.

**Figure 8 ijms-25-11600-f008:**
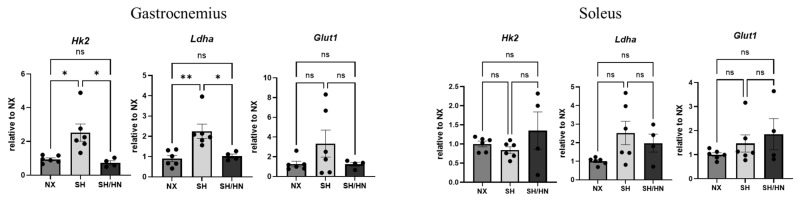
Glycolytic gene expression in the skeletal muscles of sea level normoxic (NX), Sugen/Hypoxic (SH), and Honokiol+NAD treated (SH/HN) rats. mRNA was isolated from gastrocnemius and soleus tissues of sea-level normoxic control rats (NX), Sugen/Hypoxia (SH) induced pulmonary hypertensive rats, and SH plus Honokiol+NAD treated rats (SH/HN). Real-time RT-PCR was performed to examine gene expression. Data was calculated as the fold changed relative to NX control and shown as mean ± sem. * *p* < 0.05, ** *p* < 0.01. ns, not statistically different.

**Figure 9 ijms-25-11600-f009:**
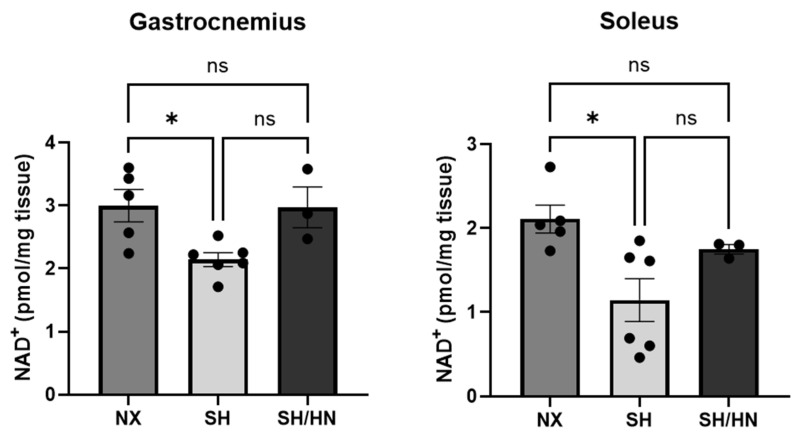
The concentrations of nicotinamide adenine dinucleotide (NAD^+^) in skeletal muscles are decreased in Sugen/Hypoxic PH rats. Honokiol and NAD treatment did not restore NAD levels. NAD^+^ concentrations were measured and calculated with gastrocnemius and soleus tissues of sea level normoxic control rats (NX), Sugen/Hypoxia (SH) induced pulmonary hypertensive rats, and SH plus Honokiol+NAD treated rats (SH/HN) using NAD/NADH colorimetric assay kit. Data is presented as mean ± sem, * *p* < 0.05. ns, not statistically different.

## Data Availability

Data are contained within the article and Appendix A.

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
