# Peer review of "Honokiol and Nicotinamide Adenine Dinucleotide Improve Exercise Endurance in Pulmonary Hypertensive Rats Through Increasing SIRT3 Function in Skeletal Muscle"

_ijms, 2024, doi:10.3390/ijms252111600_

Round 1

Reviewer 1 Report

Comments and Suggestions for Authors

This article by Li and colleagues studies the therapeutic potential of activating SIRT3 in pulmonary hypertension (PH) using Honokiol and NAD in a rat model. The researchers confirmed the classical hallmarks of PH in the model, and SIRT3 deficiency in skeletal muscle, reducing exercise capacity. After the treatment with Honokiol and NAD, they improved the exercise endurance of the animals by restoring SIRT3 levels in skeletal muscles, reducing muscle atrophy, and enhancing mitochondrial function, independent of cardiopulmonary hemodynamic changes. In conclusion, they suggest that targeting skeletal muscle dysfunction could be a promising approach to improve exercise capacity and quality of life in PH patients.

Overall, the article is well-written, the results are presented clearly and they support the conclusions. Also, the limitations are explained in detail. My only concern is that I was expecting a bit more research on the mechanism at the mitochondrial level.

Minor:

-              Lines 122-123 and 341-342 have an overstatement and I would suggest not trying to oversell the results.

Author Response

Thank you for your letter and for the opportunity to provide revisions to our manuscript. We appreciated the comments from you and all reviewers regarding our manuscript.

We addressed all the comments from three reviewers to improve our manuscript. Please see the point-by-point answers in the attached “Response to Reviewers’ comments”.

Reviewer 2 Report

Comments and Suggestions for Authors

The current manuscript is a well-structured study that investigates the effects of Honokiol and Nicotinamide Adenine Dinucleotide (NAD) on exercise endurance in a rat model of pulmonary hypertension (PH). Pulmonary hypertension impairs exercise capacity, which is influenced by multiple systems, including the lungs, right ventricle (RV), and skeletal muscles. This study focuses on the role of mitochondrial dysfunction and skeletal muscle abnormalities in PH. Results show that treatment with Honokiol and NAD improves exercise endurance by enhancing SIRT3 activity in skeletal muscles, reducing muscle atrophy, and improving mitochondrial function, independent of changes in RV and pulmonary hemodynamics. This highlights a potential therapeutic approach to improve exercise capacity in PH patients. 

I have some comment:

1. At the end of the introduction, the aim should be clear without referring to any results or conclusions.

2. Real time PCR was used to assess the expression of which genes in which tissues? This should be specified clearly in the methods.

3. The methods of PCR analysis should be written in details. Which kits were used? Which instruments? Which conditions?!!

4. Statistical analysis is written in a very confusing way. It should be rewritten in clear, direct and non-repeated sentences.

5. Were the data expressed as mean and SD or SEM? Both are mentioned in the paragraph of statistical analysis!!!!

6. All results of PCR analysis should be included in the manuscript (not as supplementary data).

7. Similarly, the results of the concentration of NAD should be also included in the manuscript (not supplementary).

8. The discussion is very long and contains mostly irrelevant data. For example, lines 270-276 should be omitted.

9. Lines 287 - 302 contain a comparison with a previous study that is not required here.

10. What is the relation of the long paragraph (lines 303 - 331) with the current study?!!

Author Response

(The authors gave the same response as above.)

Reviewer 3 Report

Comments and Suggestions for Authors

Greetings to the authors, after reading your manuscript my comments are as follows. 

In the introduction the authors should give more details about underlying causes that may lead to pulmonary hypertension they may refer to 10.3390/jcm10225272 and doi: 10.1183/09059180.00009011

The authors should make it clear what honokiol is and what studies have been made on it so far as well as it's relevance and demonstrated scientific benefits.

The same applies for NAD, the authors should also search and mention previous results from literature regarding these 2 studied elements, otherwise it seems like they are randomly testing substances, especially in the case of Honokiol. 

Not only that but the authors should also mention current therapies which have been proven to be effective in controlling PH. While indeed your study focuses on improving exercise capacity, this event naturally goes hand in hand with managing PH, as a right ventricle with reduced function cannot provide the necessary output regardless of how well the peripheral muscles adapt.

In the materials and methods section the authors must mention how many rats were used to conduct and elaborate on the study lot. 

"The treatment improved RV adaptation to the 341 increased PA pressure without altering hemodynamic parameters."  What did the authors mean by this line ? as the RV adaptation should have improved hemodynamic parameters not alter them.

The manuscript is well written and the study is interesting, however it's main limitation are the fact that it studies a type of traditionalist treatment which has not been demonstrated it's effectiveness in any clinical trials. Not only that but the fac that PH is such a complex disease that subjects' exercise tolerance depends on the state of the lungs, heart as well as peripheral muscles. From a molecular standpoint the study is interesting. 

Author Response

(The authors gave the same response as above.)

Round 2

Reviewer 2 Report

Comments and Suggestions for Authors

The authors have made the required modifications.

Reviewer 3 Report

Comments and Suggestions for Authors

The authors have adjusted the manuscript according to my recommendations.